# Understanding the complexity of socioeconomic disparities in smoking prevalence in Sweden: a cross-sectional study applying intersectionality theory

Sten Axelsson Fisk [1], Martin Lindström,[2,3] Raquel Perez-Vicente,[1] Juan Merlo [1,3]

¹Unit for Social Epidemiology, Department of Clinical Sciences, Lund University, Malmö, Sweden
²Unit for Social Medicine and Health Policy, Department of Clinical Sciences, Lund University, Malmö, Sweden
³Center for Primary Health Care Research, Region Skåne, Malmö, Sweden

**Correspondence to**
Dr Sten Axelsson Fisk;
sten.axelsson_fisk@med.lu.se

## ABSTRACT

**Objectives** Socioeconomic disparities in smoking prevalence remain a challenge to public health. The objective of this study was to present a simple methodology that displays intersectional patterns of smoking and quantify heterogeneities within groups to avoid inappropriate and potentially stigmatising conclusions exclusively based on group averages.

**Setting** This is a cross-sectional observational study based on data from the National Health Surveys for Sweden (2004–2016 and 2018) including 136 301 individuals. We excluded people under 30 years of age, or missing information on education, household composition or smoking habits. The final sample consisted on 110 044 individuals or 80.7% of the original sample.

**Outcome** Applying intersectional analysis of individual heterogeneity and discriminatory accuracy (AIHDA), we investigated the risk of self-reported smoking across 72 intersectional strata defined by age, gender, educational achievement, migration status and household composition.

**Results** The distribution of smoking habit risk in the population was very heterogeneous. For instance, immigrant men aged 30–44 with low educational achievement that lived alone had a prevalence of smoking of 54% (95% CI 44% to 64%), around nine times higher than native women aged 65–84 with high educational achievement and living with other(s) that had a prevalence of 6% (95% CI 5% to 7%). The discriminatory accuracy of the information was moderate.

**Conclusion** A more detailed, intersectional mapping of the socioeconomic and demographic disparities of smoking can assist in public health management aiming to eliminate this unhealthy habit from the community. Intersectionality theory together with AIHDA provides information that can guide resource allocation according to the concept proportionate universalism.

## INTRODUCTION

A higher prevalence of smoking among individuals with low socioeconomic position (SEP) compared with higher SEP has been reported in several studies in Sweden[1] and globally.[2–5] The higher prevalence results both from higher rates of initiation[6] and lower rates of successful smoking cessation.[7]

### Strengths and limitations of this study

► We present an intersectional approach to study the multidimensional socioeconomic disparities in smoking prevalence in Sweden.
► In addition to differences between averages of intersectional strata, we quantify individual heterogeneities around those averages by presenting measurements of discriminatory accuracy.
► Our method is simpler but share crucial advantages with multilevel analysis of individual heterogeneity and discriminatory accuracy (AIHDA), such as improved health mapping and assessment of intersectional interaction.
► We use pooled data from Swedish National Health Survey with participation rates spanning from 60.8% 2004 to 42.1% 2018.
► AIHDA is a suitable tool to inform whether interventions to reduce socioeconomic health disparities should be universal or target-specific groups.

In addition to this, other factors like country of birth,[8] household composition,[9] age and gender influence the probability of smoking.[10] Overall, socioeconomic determinants of smoking are multidimensional but few studies have empirically confronted this heterogeneity using an intersectional perspective.[11–15]

### Intersectionality theory, proportionate universalism and the analysis of individual heterogeneity and discriminatory accuracy

Structural interventions including raised tobacco taxes and smoking-free zones can reduce smoking prevalence,[16] most among people with low SEP.[17] In UK, healthcare-based smoking cessation aid has reduced disparities in smoking rates between privileged and socioeconomically deprived areas, although this effect was modest.[18] However, a review of the efficacy of non-healthcare interventions targeting behavioural factors

among people with low education[19] concludes that there is a lack of evidence that such interventions oriented towards individual determinants of health are efficient when it comes to reducing socioeconomic disparities in smoking.[20] Marmot and Bell claim[21] that interventions to reduce socioeconomic health disparities need to address all levels of society and not only those who are worst off. They argue that an efficient approach may be proportionate universalism[21 22] where interventions are universal, that is, directed towards the whole population (such as tobacco taxes, smoking bans in public) but proportionately more intense among population subgroups with augmented needs where targeted interventions can be launched (ie, information campaigns in specific neighbourhoods or populations such as pregnant women). However, as argued elsewhere[22–24] successful and efficient implementation of proportionate universalism requires development and application of appropriate theories and epidemiological methodologies.

Intersectionality theory is a critical social theory[25] that stresses the need for simultaneous consideration of different social dimensions such as racialised identity, gender and class in order to properly understand the social context acting on individuals. According to intersectionality theory, the social reality is shaped by overlapping systems of oppression that influence distribution of resources and power in society.

The inclusion of intersectionality in epidemiology and public health has been promoted by several scholars.[26–29] A direct consequence of this approach in quantitative analyses is the study of multiple intersectional strata defined by combinations of different social dimensions, since the effect of each social dimension on an individual is intrinsically dependent on other social identities of that person. This contrasts with the common approach considering one social dimension at the time. Thereby, the intersectional approach may enrich public health research by providing an improved mapping of socioeconomic health disparities.[26 30] Such socioeconomic heterogeneity can be analysed by quantifying differences between intersectional strata averages. However, we[23 28 29 31 32] and other scholars[33–35] stress the added relevance of simultaneously quantifying the discriminatory accuracy (DA) of the intersectional categorisation for specific outcomes. An intersectional map combined with information on its DA provides an improved picture of the socioeconomic heterogeneity existing in the society. This approach can be used to inform interventions according to the concept of proportionate universalism. The extent to which a universal intervention needs to be proportional can be evaluated by the DA of the intersectional strata. A low DA suggests the need for universal interventions while a high DA supports more selective interventions. This idea aligns with the distinctions made by McCall between anticategorical, and intercategorical intersectional approaches.[36] According to the anticategorical intersectionality, the categorisations adopted in quantitative research are simplified and contribute to stereotypes

and perpetuations of inequalities. The intercategorical intersectionality, on the other hand, accepts categorisations since they can be useful in the study of intersectional inequities. The finding of a low DA would support the anticategorical standpoint that the categorisations lack relevance for the studied outcome. If the DA is high, this would rather support the intercategorical standpoint that intersectional matrix provides worthy information. A moderate DA does not give full support to neither the anticategorical nor intercategorical intersectionality.

Adopting a quantitative perspective, in the present study, we aim to illustrate how a more precise intersectional categorisation combined with analysis of individual heterogeneity and DA (AIHDA) improves our understanding of smoking prevalence and facilitates the application of proportionate universalism.

## METHODS

### Study population

In this cross-sectional observational study, we used data from all the 14 National Health Surveys (NHS) for Sweden for the years 2004–2016 and 2018 (https://www.folkhalsomyndigheten.se/the-public-health-agency-of-sweden/public-health-reporting/). The NHS is an ongoing collaborative project between the Public Health Agency of Sweden and the Swedish Association of Local Authorities and Regions. The NHS record self-reported information on health, lifestyle and living conditions. The study has been conducted annually between 2004 and 2016 and comprised a random sample of 20 000 individuals aged 16–84 years. After 2016 the survey is conducted biannually but with a random sample of 40 000 individuals. Response rates span from 60.8% 2004 to 42.1% 2018. Using a unique personal identification number, the Swedish authorities linked the sample surveys to national register administered at Statistics Sweden to obtain demographical and socioeconomic information.

For our study, we pooled the data from the last 14 surveys, which rendered a sample of 136 301 individuals. Thereafter, we excluded people younger than 30 years. The lower age limit of 30 years was chosen since most individuals in Sweden that will complete a 3-year education after high school do so before this age[37] and educational status was the indicator of SEP chosen in this study. We also excluded people with missing information on education, household composition or smoking habits. The final sample consisted on 110 044 individuals or 80.7% of the original sample (figure 1).

### Patient and public involvement

All data from NHS provided to researchers is anonymised, so study participants cannot be identified. The study participants were not involved in the research process.

### Assessment of variables

Smoking status was assessed based on the answer to the question 'Do you smoke?', if the person answered 'yes'

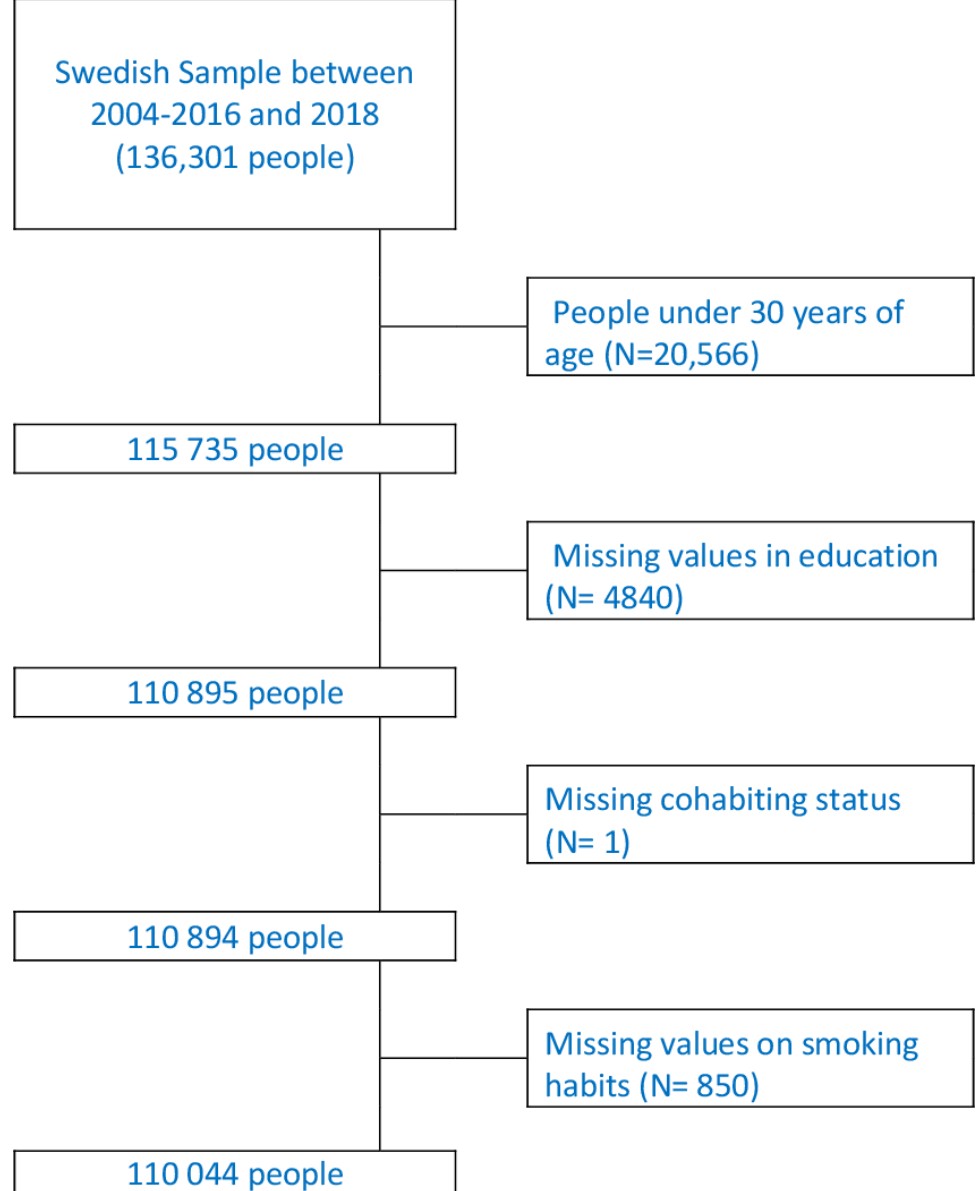

**Figure 1** Flow chart showing the selection of the study population.

or 'yes, sometimes', the individual was categorised as a smoker, if the respondent answered 'no' the individual was considered a non-smoker.

We categorised age into three groups: 30–44, 45–64 and 65–84 year-old. We classified gender as a binary variable distinguishing between men and women as more specific information on gender was not available in the questionnaire. We classified educational achievement into three categories, as low if the respondent had not completed 3 years of high school education, as middle if they had high school education but less than 3 years of education after high school and high if the respondent had at least 3 years of education after high school. Throughout 2008–2016 respondents were asked 'with whom do you share household?', we defined household composition as living alone if the respondent answered 'with no one', otherwise as living with other(s). In 2018 that question was not asked so individuals were defined in the same way

according to the linked information provided by Statics Sweden. We classified migration status as native (ie, born in Sweden) or immigrant.

As a way of operationalising intersectional contexts, we created 72 strata by combining the three categories of age, the two of gender, the three of educational achievement, the two of migration status and the two categories of household composition. We used 30–45 years old native men living with other(s) and with high educational achievement as the reference in the comparisons, as this group was assumed to occupy the position of greatest structural privilege. This choice was based on unidimensional assumptions of structural privilege for young compared with old,[38] men compared with women, high SEP compared with low SEP,[39] natives compared with immigrants[40] and those living with other(s) compared with people living alone.[41] We also included the survey year of the participants using 2018 as reference in all comparisons.

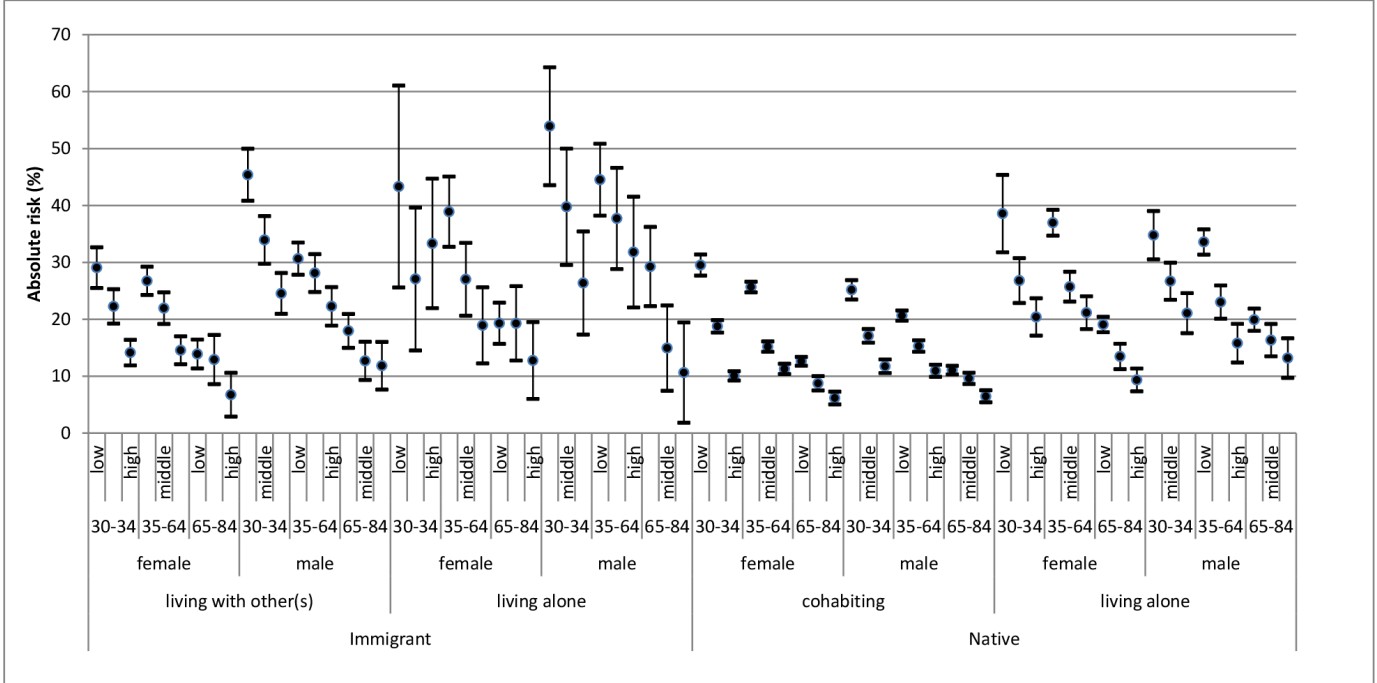

**Figure 2** Absolute risk (ie, prevalence) and 95% CIs of smoking in different intersectional strata according the National health survey in Sweden between 2004 and 2018.

## Statistical analyses

The first step in our analysis was to obtain the trends in smoking prevalence and the trends in socioeconomic and demographic gradients in smoking between 2004 and 2018 (see online supplemental material 1). Thereafter, we performed a stratified analysis aimed to provide a detailed map of the prevalence (ie, absolute risk) and 95% CIs of smoking across the intersectional strata. This stratification allows comparing the prevalence of smoking in different strata without any reference (figure 2).

Thereafter, we performed seven consecutive regression analyses, modelling smoking as the dependent variable and survey year as well as the different demographic and socioeconomic dimensions alone and in combination as explanatory variables. The use of logistic regression to obtain ORs is common but the OR is a good estimation of the relative risk only when the prevalence of the outcome is very small (rare event assumption).[42] Therefore, for the analysis, rather than logistic regression to obtain ORs, we used Cox proportional hazards regression with a constant follow-up time equal to one to obtain prevalence ratios (PR)[43] with 95% CI.

Model 1 included only survey year, model 2 added age, model 3 added gender, model 4 added educational achievement, model 5 added migration status and model 6 added household composition and thus included all the variables that defined the intersectional strata. Finally, the intersectional model 7 included the same variables as model 6 but in the form of a multicategorical variable with 72 intersectional strata. Here, we used the 30–45 years

old, native men living with other(s) and with high educational achievement as the reference in the comparison.

For each model, we quantified its DA by means of the area under the receiver operator characteristics curve (AUC).[44] The AUC measures the accuracy of the information provided by the variables in the model for discriminating individuals who smoke from those who do not. The AUC takes a value between 0.5 and 1, where 1 indicates perfect discrimination and 0.5 means that the studied variables have no DA at all. The AUC can even be used to qualify the size of the intersectional differences. Rather than evaluating the absolute risk differences between strata, using the AUC we assess the overlapping of the individual risk predictions (based on the intersectional strata) between smokers and non-smokers.

There is no fully established practical guideline for the interpretation of the size of the AUC as a measure of DA when analysing intersectional inequalities. However, based on the cut-off values provided by Hosmer and Lemeshow[45] but using more neutral denominations we qualify intersectional inequalities according to the DA as (1) 'absent or very small' (AUC=0.5–0.6), (2) 'moderate' (AUC >0.6–≤0.7), (3) 'large' (AUC >0.7–≤0.8) and (4) 'very large' (AUC >0.8). Evaluating intersectional differences using only strata prevalence is insufficient as it does not consider any overlapping between the strata. Therefore, the AUC provides fundamental information for evaluation of group differences.[46]

We further calculated the incremental change in the AUC value (Δ-AUC) between the models. The Δ-AUC

quantifies the improvement in the DA obtained by a model, in relation to the previous model.[24] The categorical intersectional variable in model 7 allows for the capturing of interaction of effects. If any such interaction exists, the DA of model 7 will increase in comparison with model 6 and the Δ-AUC will thus be positive.

We used STATA V.15.1 and IBM SPSS V.25 for PC to perform all statistical analyses.

## RESULTS

Over the whole study period, the prevalence of smoking was 18%. The visual analysis of the trends indicated that the prevalence of smoking monotonically decreased in Sweden from 25.0% in 2004 to around 11.1% in 2018. While sex-differences were small throughout the period and the sex-category with highest smoking prevalence changed, we observed consistent differences between groups defined by age, country of birth, educational achievement and household composition. In absolute terms, the gaps between subgroups were static except for differences between age categories that narrowed in later years (see online supplemental material 1).

Table 1 presents the prevalence of smokers and non-smokers across the included socioeconomic and demographic variables as well as across survey years. It indicates that the prevalence of smoking was higher in individuals aged 45–64 years (20.6%) than in both younger (19.8%) and older people (12.4%). Women and men had similar prevalence of smoking (17.9% vs 17.8%). As expected, smoking was more common among people with low (21.7%) and medium (17.0%) educational achievement compared with people with high educational achievement (11.9%). The prevalence of smoking was higher among immigrants (23.9%) than among natives (17.0%) and the same was true for individuals living alone (24.1%) compared with those who were living with other(s) (16.5%).

Figure 2 shows the prevalence of smoking across the intersectional strata. We observed the highest prevalence (54%) among 30–44 years old immigrant men with low educational achievement and living alone, and the lowest prevalence (6%) among 65–84 years old native women with high educational achievement and living together. The reference stratum (ie, 30–45 years old, native men living with other(s) and with high educational achievement) used in the relative comparisons (table 2) presented a smoking prevalence of about 12%.

The table 3 informs that the PR of smoking decreases with age, being lowest in the old population. This age gradient is clear after adjustment for the other variables in the model 6. Low educational achievement, being immigrant and living alone was associated with a higher smoking risk. However, there were no age-adjusted gender differences. The AUC in the model including only survey year was 0.58. In the age adjusted model 2, the AUC was 0.60 and it did not increase when gender was included in model 3. The AUC increased by 0.04

**Table 1** Distribution (prevalence) of smokers across categories of age, gender, education, migration and household composition in the 110 044 participants in the Swedish National Health Surveys (2004–2018)

| | Non-smokers (%) | Smokers (%) |
|---|---|---|
| 30–44 | 22 799 (80.23) | 5618 (19.77) |
| 45–64 | 38 024 (79.41) | 9862 (20.59) |
| 65–84 | 29 575 (87.65) | 4166 (12.35) |
| Female | 48 782 (82.08) | 10.653 (17.92) |
| Male | 41 616 (82.23) | 8993 (17.77) |
| Low | 38 791 (78.32) | 10 738 (21.68) |
| Middle | 27 716 (83.02) | 5670 (16.98) |
| High | 23 891 (88.06) | 3238 (11.94) |
| Immigrant | 10 410 (76.07) | 3274 (23.93) |
| Native | 79 988 (83.01) | 16 372 (16.99) |
| Living with other(s) | 75 625 (83.48) | 14 964 (16.52) |
| Living Alone | 14 773 (75.93) | 4682 (24.07) |
| 2004 | 6803 (75.03) | 2264 (24.97) |
| 2005 | 3339 (75.90) | 1060 (24.10) |
| 2006 | 3450 (77.62) | 995 (22.38) |
| 2007 | 3272 (77.81) | 933 (22.19) |
| 2008 | 6525 (79.07) | 1727 (20.93) |
| 2009 | 6123 (79.22) | 1606 (20.78) |
| 2010 | 6718 (80.59) | 1618 (19.41) |
| 2011 | 6760 (82.56) | 1428 (17.44) |
| 2012 | 6893 (82.68) | 1444 (17.32) |
| 2013 | 6770 (83.10) | 1377 (16.90) |
| 2014 | 6845 (83.74) | 1329 (16.26) |
| 2015 | 6978 (84.21) | 1308 (15.79) |
| 2016 | 7086 (88.13) | 954 (11.87) |
| 2018 | 12 836 (88.90) | 1603 (11.10) |

Values are number (and percentage) of individuals.

units when including education. It did not increase when adding migration status but further increased by 0.01 units when including household composition. The AUC of intersectional model 7 was 0.66, with 95% CI overlapping the AUC of model 6 indicating no conclusive intersectional interaction.

Table 2 shows the 10 strata with the lowest and the 10 strata with the highest PRs of smoking using the strata of young native men with high educational achievement and living with other(s) as reference. The lowest PR=0.55 was observed in older native women with high educational achievement and living with other(s) and the highest PR=4.45 was observed in young immigrant men with low educational achievement and living alone. When comparing with the reference stratum of native young men with high educational achievement and living with other(s), we observed that low educational achievement, being immigrant and living alone were, respectively,

**Table 2** Results from the intersectional model 7 indicating the 10 strata with lowest and the 10 strata with highest prevalence ratios (PR) with 95% CIs of smoking across intersectional strata in the Swedish population using the stratum of young, native, men with high education that were living with other(s) (LWO) as reference in the comparisons

| Age | Gender | Educational achievement | Migration status | Household composition | PR (95% CI) |
|-----|--------|------------------------|------------------|----------------------|-------------|
| 65–84 | Female | High | Native | LWO | 0.55 (0.45 to 0.69) |
| 65–84 | Male | High | Native | LWO | 0.58 (0.48 to 0.71) |
| 65–84 | Female | High | Immigrant | LWO | 0.61 (0.33 to 1.11) |
| 65–84 | Female | Middle | Native | LWO | 0.80 (0.66 to 0.96) |
| 65–84 | Female | High | Native | Living alone | 0.83 (0.64 to 1.06) |
| 65–84 | Male | Middle | Native | LWO | 0.85 (0.73 to 0.99) |
| 30–44 | Female | High | Native | LWO | 0.86 (0.74 to 0.98) |
| 65–84 | Male | High | Immigrant | Living alone | 0.91 (0.38 to 2.21) |
| 45–64 | Male | High | Native | LWO | 0.92 (0.8 to 1.07) |
| 65–84 | Male | Low | Native | LWO | 0.96 (0.84 to 1.11) |
| 30–44 | Male | High | Native | LWO | Reference |
| 30–44 | Female | High | Immigrant | Living alone | 2.87 (1.86 to 4.42) |
| 30–44 | Female | Low | Native | Living alone | 2.95 (2.29 to 3.78) |
| 45–64 | Female | Low | Native | Living alone | 2.99 (2.61 to 3.41) |
| 45–64 | Male | Middle | Immigrant | Living alone | 3.10 (2.26 to 4.26) |
| 45–64 | Female | Low | Immigrant | Living alone | 3.22 (2.56 to 4.06) |
| 30–44 | Male | Middle | Immigrant | Living alone | 3.33 (2.35 to 4.71) |
| 30–44 | Female | Low | Immigrant | Living alone | 3.41 (1.96 to 5.94) |
| 45–64 | Male | Low | Immigrant | Living alone | 3.61 (2.90 to 4.50) |
| 30–44 | Male | Low | Immigrant | LWO | 3.66 (3.07 to 4.35) |
| 30–44 | Male | Low | Immigrant | Living alone | 4.45 (3.29 to 6.03) |
| AUC | | | | | 0.66 (0.65 to 0.66) |
| ΔAUC compared with model 6 | | | | | 0.01 |

AUC, area under the curve.

present in 7, 8 and 9 of the 10 strata with the highest risk of smoking (see online supplemental material 2 for the complete list of PR values).

## DISCUSSION
### Main findings
Our study provides an improved mapping of the distribution of the smoking habit in Sweden compared with unidimensional analyses. Rather than focusing on single socioeconomic and demographical variables, we use an intersectional AIHDA analysis that uncovers the socioeconomic and demographical heterogeneity existing in the country. We also applied the AUC to obtain information on the accuracy of the intersectional grouping for identifying individuals according to their smoking status. We found a moderate AUC=0.66, which indicates that individual risk of smoking considerably overlaps between the intersectional strata and that neither the anticategorical nor the intercategorical intersectionality approaches are fully supported. We found that the stratum-specific risks were due to the main effects of the different variables used

to define the intersectional strata without any conclusive interactive component.

We found intersectional strata with a rather high prevalence of smoking. For instance, the prevalence of smoking in young immigrant men with low educational achievement and living alone was 54%. Interestingly, while high educational achievement generally prevents smoking, young immigrant women that lived alone had a PR of 2.87 (95% CI 1.86 to 4.42) despite their high educational achievement. This indicates that the protective effect of high education may depend on other variables such as migration status and gender. Our finding could hypothetically reflect both smoking culture in the country of birth of the individual or that discrimination on the basis of gender or migration status may contribute to making education a poorer indicator of SEP in this group.

### Relation to previous studies
In spite of the use of different definitions and measurements of smoking habits as well as the use of different indicators of SEP, many previous publications have shown the existence of socioeconomic, ethnic and

**Table 3** Prevalence ratios (PR) and 95% CI, of smoking among people aged 30–84 included in the National Health Surveys between 2004 and 2018 in relation to survey year, age, gender, education, migration statis and household composition

| | | Model 1 | Model 2 | Model 3 | Model 4 | Model 5 | Model 6 |
|---|---|---|---|---|---|---|---|
| Year | 2004 | 2.25 (2.11–2.40) | 2.07 (1.95–2.21) | 2.08 (1.95–2.21) | 1.85 (1.74–1.98) | 1.88 (1.76–2.00) | 1.84 (1.73–1.97) |
| | 2005 | 2.17 (2.01–2.35) | 2.01 (1.86–2.17) | 2.01 (1.86.2.17) | 1.83 (1.69–1.98) | 1.85 (1.71–2.00) | 1.83 (1.69–1.98) |
| | 2006 | 2.02 (1.86–2.18) | 1.87 (1.72–2.02) | 1.87 (1.72–2.02) | 1.70 (1.57–1.84) | 1.71 (1.58–1.85) | 1.68 (1.55–1.82) |
| | 2007 | 2.00 (1.84–2.17) | 1.86 (1.71–2.01) | 1.86 (1.71–2.01) | 1.71 (1.57–1.85) | 1.65 (1.54–1.77) | 1.70 (1.56–1.84) |
| | 2008 | 1.89 (1.76–2.02) | 1.76 (1.64–1.88) | 1.76 (1.64–1.88) | 1.64 (1.53–1.75) | 1.66 (1.55–1.78) | 1.63 (1.52–1.74) |
| | 2009 | 1.87 (1.75–2.01) | 1.75 (1.64–1.89) | 1.76 (1.64–1.89) | 1.65 (1.54–1.77) | 1.62 (1.51–1.73) | 1.63 (1.52–1.75) |
| | 2010 | 1.75 (1.63–1.87) | 1.70 (1.58–1.82) | 1.70 (1.58–1.82) | 1.61 (1.50–1.72) | 1.46 (1.36–1.57) | 1.59 (1.48–1.70) |
| | 2011 | 1.57 (1.46–1.69) | 1.53 (1.43–1.64) | 1.53 (1.43–1.64) | 1.45 (1.35–1.56) | 1.47 (1.37–1.58) | 1.43 (1.33–1.54) |
| | 2012 | 1.56 (1.45–1.68) | 1.53 (1.42–1.64) | 1.53 (1.42–1.64) | 1.47 (1.37–1.57) | 1.45 (1.35–1.56) | 1.44 (1.34–1.55) |
| | 2013 | 1.52 (1.42–1.64) | 1.49 (1.39–1.60) | 1.49 (1.39–1.60) | 1.45 (1.35–1.55) | 1.45 (1.35–1.56) | 1.42 (1.32–1.52) |
| | 2014 | 1.47 (1.36–1.75) | 1.45 (1.35–1.56) | 1.45 (1.35–1.56) | 1.41 (1.31–1.52) | 1.42 (1.32–1.53) | 1.39 (1.30–1.50) |
| | 2015 | 1.42 (1.32–1.53) | 1.40 (1.30–1.51) | 1.40 (1.30–1.51) | 1.37 (1.27–1.47) | 1.38 (1.28–1.48) | 1.35 (1.26–1.46) |
| | 2016 | 1.07 (0.99–1.16) | 1.06 (0.97–1.14) | 1.06 (0.97–1.14) | 1.04 (0.96–1.13) | 1.05 (0.97–1.13) | 1.03 (0.95–1.11) |
| | 2018 | Reference | Reference | Reference | Reference | Reference | Reference |
| Age | 30–44 | | Reference | Reference | Reference | Reference | Reference |
| | 45–64 | | 1.06 (1.03–1.10) | 1.06 (1.03–1.10) | 0.94 (0.91–0.97) | 0.94 (0.91–0.98) | 0.93 (0.90–0.96) |
| | 65–84 | | 0.68 (0.65–0.71) | 0.68 (0.65–0.71) | 0.56 (0.54–0.58) | 0.57 (0.55–0.59) | 0.53 (0.51–0.56) |
| Gender | Male | | | Reference | Reference | Reference | Reference |
| | Female | | | 1.00 (0.97–1.02) | 1.02 (0.99–1.05) | 1.02 (0.99–1.05) | 1.01 (0.98–1.04) |
| Education | Low | | | | 1.96 (1.88–2.04) | 1.96 (1.88–2.04) | 1.93 (1.86–2.01) |
| | Middle | | | | 1.42 (1.36–1.48) | 1.43 (1.37–1.49) | 1.42 (1.36–1.49) |
| | High | | | | Reference | Reference | Reference |
| Born in Sweden | Native | | | | | Reference | Reference |
| | Immigrant | | | | | 1.39 (1.34–1.45) | 1.39 (1.24–1.44) |
| Living alone | Living alone | | | | | | 1.53 (1.48–1.58) |
| | living with other(s) | | | | | | Reference |
| AUC | | 0.58 (0.58–0.59) | 0.60 (0.60–0.61) | 0.60 (0.60–0.61) | 0.64 (0.63–0.64) | 0.64 (0.64–0.65) | 0.65 (0.65–0.66) |
| ΔAUC | | – | 0.02 | 0.00 | 0.04 | 0.00 | 0.01 |

Model 7 includes the same variables as model 6 but as a multicategorical variable, the PRs and AUC for model 7 are presented in the table 2. AUC values with 95% CI representing the discriminatory accuracy and ΔAUC values of the models are also presented.
AUC, area under the curve.

demographical differences in smoking.[2 4 47] However, as far we known, only a few have considered the intersectional approach.[11 14 15] The heterogeneous distribution of smoking prevalence we found in Sweden is in accordance with recent intersectional research on smoking cessation in the US adult population.[15]

High education may influence smoking through both direct effects, such as increased understanding of detrimental health effects of smoking, and indirect effects such as social and material circumstances.[48] Educational achievement is the preferred indicator of SEP in previous public health reports in Sweden.[49] We performed a sensitivity analysis where we included income instead of education and the results were very similar and are provided as online supplemental material 3.

In a comparison of the relative importance of low education on smoking prevalence across age and gender groups in Denmark and Sweden, Eek et al[1] found that the effect of low education on smoking prevalence and continuation of smoking was strongest among younger women in Sweden, indicating a failure of tobacco prevention interventions to reach this group. While immigrant men were clearly overrepresented among the strata with highest prevalence of smoking, this was not the case for women. This pattern was also found by Lindström and Sundquist[8] in a study from southern Sweden showing lower rates of smoking among men born in Sweden, but higher rates of smoking among women born in Sweden compared with men and women from most other country groups. These differences were attributable to different smoking prevalence in the countries of origins of the immigrants, potentially representing different stages of the smoking transition. The distribution of smoking prevalence across age groups we found is similar to the pattern observed by Ali et al[50] in a study from southern Sweden.

### Strengths and limitations

The cross-sectional and observational character of this study prevents causal conclusions. However, the variables included in our analyses are to a little extent effected by smoking status, so the causal direction can be presumed to go from sociodemographic variables towards smoking rather than the opposite.

A weakness in our study is that the participation rates were rather low, especially during the last years. An analysis of the non-participants performed by Statistics Sweden shows that people with low income, people born outside Sweden and people living alone were less likely to be responders.[51] Therefore, if the prevalence of smoking is higher in non-participants, our analysis may have underestimated the existing socioeconomic differences. In a sensitivity analysis, we used data that had been weighted by Statistics Sweden in order to reduce skewness resulting from non-participating individuals. The variables used to perform the weighting were age, gender, educational level, country of birth, household composition and urban/rural.[52] These results were very similar, which was expected since the intersectional

variable included all weighting variables except rural/urban. Our study represents the Swedish circumstances so the AIHDA-approach should be replicated in different contexts.

A further limitation of this study is the simple categorisations of the dimensions incorporated in the intersectional matrix. Gender was binary defined which neglects the existence of numerous gender identities. Migration status was binary defined as natives and immigrants, which may hide heterogeneity in smoking prevalence. A more detailed classification with four categories (ie, Sweden, Nordic countries, Europe and Outside Europe) shows that all the categories except women born outside Europe had a higher prevalence than the individuals born in Sweden (see online supplemental material 4). The used categorisations stem in part from the information available in the survey and in part from the aim of presenting a parsimonious intersectional model that is easier to adopt in public health analyses and by the fact that several strata would be empty or contain very few individuals if the intersectional matrix was expanded.

We also performed a sensitivity analysis excluding 'sometimes smokers' from the smoker category. As expected, overall prevalence was lower, 11% compared with 18%, and intersectional disparities larger. The AUC of the intersectional model 7 was 0.70 compared with 0.66 in the main analysis. Our main results combined with the results from the sensitivity analysis reflect the existence of socioeconomic disparities not only in prevalence, but also in intensity, of smoking.[53] Our results, therefore, may underestimate the intersectional disparities in health hazards attributable to smoking.

### Implications and future studies

There is a growing body of literature focusing on how to perform quantitative intersectional research,[27 36] with the emergence of multilevel AIHDA (MAIHDA) as a recent example.[28 29 34] However, in spite of providing complementary information,[34] the fixed effects AIHDA approach we use in our study is rather accessible and share crucial advantages of the MAIHDA. First, the AIHDA provides an intersectional mapping that is more appropriate than unidimensional analyses to identify specifically vulnerable population groups in which interventions could be effective. Second, by going beyond average probabilistic measurements (ie, prevalence) and also analysing DA we get a quantification of the heterogeneity around the averages.[46] From the AIHDA, we found that the DA of our intersectional model was only moderate which indicates the necessity for universal interventions due to a large unexplained heterogeneity. However, we also identified that the three most vulnerable groups (ie, strata) included immigrant men with low education younger than 65 years. This finding suggests that special preventive measures should be directed to these groups. Furthermore, research methods that actively involves members of marginalised

groups and has the explicit purpose to result in public health improvements are developing and could be one way forward.[54]

Interventions to reduce smoking prevalence should address Social Determinants of Health (SDH) at all levels. Examples targeted directly at smoking include increased tobacco taxation, smoke-free zones and public antismoking campaigns.[55] Stigmatisation is a negative side effect of such interventions that need to be taken into account, especially for low SEP groups.[56] Qualitative intersectional research has provided important insights into how the stigma of smoking interacts with identities of low class, country of birth, being a bad mother and may be in conflict with norms of femininity.[57]

Equal access to education, housing and healthy recreation, regardless of gender, socioeconomic status, migration status and household composition, is important to reduce smoking prevalence. Therefore, institutions outside the healthcare system play an important role to redistribute resources and access to SDH,[58 59] in order to counterweight the accelerating tendency of accumulation of resources among a very rich minority that characterises modern capitalism.[60] This requires political decisions that prioritise population health aims more than market-oriented reforms that exacerbate health inequities.[61] Health politics should adopt an intersectional perspective when redistributing resources in order to reduce the complex disparities in smoking revealed in this study.

## CONCLUSIONS

Compared with studies focused on single variables, the intersectional AIHDA offers a better mapping of the socio-economic and demographical distribution of smoking in Sweden. However, the moderate DA found in the AIHDA analysis suggested the existence of substantial unexplained heterogeneity in smoking risk within the different intersectional strata defined by age, gender, education, household composition and migration status. An intersectional AIHDA approach is necessary to understand the existing socioeconomic and demographic complexity influencing smoking behaviour. Future studies should identify preventive measures that are guided by proportionate universalism to find practical ways forwards to reduce intersectional disparities in smoking prevalence.

**Acknowledgements** The authors wish to thank the County Council in Region Skåne for providing financial and administrative support to this study.

**Contributors** SAF and JM had the original idea of the study. SAF wrote the initial version of the paper. SAF, JM, RP-V and ML participated in the design, analysis, interpretation of data and drafting of the article. All authors approved both the original and revised versions to be published. JM is the guarantor of the article.

**Funding** This work was supported by grants to Juan Merlo (PI) from the Swedish Research Council (Vetenskapsrådet) for the project 'Multilevel Analyses of Individual Heterogeneity: innovative concepts and methodological approaches in Public Health and Social Epidemiology' (https://www.swecris.se/betasearch/details/project/201701321VR?lang=en. roject id: 2017-01321.) The study was also supported by a grant for advanced doctoral studies from the Faculty of Medicine at Lund University (Dnr 2020/1125, SAF).

**Disclaimer** The funders had no role in study design, data collection and analysis, decision to publish, or preparation of the manuscript. We acknowledge the staff and participants of all the national health surveys that has made this study possible.

**Competing interests** None declared.

**Patient consent for publication** Not required.

**Ethics approval** The present investigation was approved by the Swedish Ethical Review Authority (Dnr: 2019–01793) and the data safety committee at the Public Health Agency of Sweden.

**Provenance and peer review** Not commissioned; externally peer reviewed.

**Data availability statement** Data may be obtained from a third party and are not publicly available. The original databases from the Public Health Agency of Sweden used in this study are protected by strict rules of confidentiality that cover sensitive individual information including data regarding health and health behaviours. The data can be made available for research after review by a regional Ethics Committee and the Public Healt Agency's own safety committee.

**ORCID iDs**
Sten Axelsson Fisk http://orcid.org/0000-0002-8110-8376
Juan Merlo http://orcid.org/0000-0001-8379-9708

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
