## [Reviewer comments · BMJ Open]

ARTICLE DETAILS

TITLE (PROVISIONAL)	Understanding the complexity of socioeconomic disparities in smoking prevalence in Sweden: a cross-sectional study applying intersectionality theory
AUTHORS	Axelsson Fisk, Sten; Lindstrom, Martin; Perez-Vicente, Raquel; Merlo, Juan

VERSION 1 – REVIEW

REVIEWER	Rosemary Hiscock University of Bath, UK
REVIEW RETURNED	17-Sep-2020

GENERAL COMMENTS	This study breaks the Swedish population into different sociodemographic groups and looks at changes in smoking prevalence over time. The study finds that not all low SES groups have high prevalence and not all high SES groups have low prevalence I think this point could be made clearer by adding a table/figure comparing smoking rates by low, mid, high education for different sub groups. Throughout: “civil status” should be changed to “household composition” – otherwise could be about citizenship or marital status. Also ‘cohabiting’ normally refers to with a partner so please change to ‘living with other(s)’ P6 line 32 to 39 taxes and smoke free are more effective than smoking cessation programs https://link.springer.com/article/10.1007/s12254-019-0485-6 I also disagree that it necessarily widens inequalities – if it is targeted to areas with high smoking rates (low SES) then it can reduce inequalities https://www.ncbi.nlm.nih.gov/pmc/articles/PMC2807194/ P9 line 6 why exclude those under 30 years old? P9 line 25 I think you should make clearer here that education is a measure of SES and the other variables are demographic characteristics P9 Line 56 add ‘linked’ before ‘information’ P10 line 35 Is this not just a forward stepwise? P10 line 54 Why enter in that order? P12 line 27 “such differences were similar” would it be better to say “the gaps between subgroups were static”? - however looking at the graphs S1 – the gap for age categories declines over time, the gender with the highest smoking rate changes but there is little difference by gender. Would it be possible to include confidence intervals? Partos et al’s figure 2 shows a good way of including confidence intervals using shading
---

	https://www.ncbi.nlm.nih.gov/pmc/articles/PMC5934656/ and here figure 11 https://www.ncbi.nlm.nih.gov/books/NBK555625/ Discussion P16 to 17 A big limitation and area for future work is that you lumped all foreign born participants together. Some ethnic groups/countries will have higher smoking rates than those born in Swedes and others will have lower (particularly if differentiated by gender). It would be helpful to include the largest ethnic minorities in Sweden and whether the smoking rates of these countries are higher than Sweden. Also the ethnic make up of your oldest age group is possibly different to the ethnic composition of your youngest age group? Figure 2 Please turn to landscape so that all the categories on the x axis can be seen
--	---

REVIEWER	Jeevitha Mariapun Clinical School Johor Bahru, Jeffrey Cheah School of Medicine and Health Sciences, Monash University Malaysia, Malaysia
REVIEW RETURNED	24-Oct-2020

GENERAL COMMENTS	This is an interesting piece of work that uses a simplified methodology to describe intersectional patterns of smoking and quantify heterogeneities within groups in a population. There are a few comments as follows: (1) It is stated that the original sample has 136 301 individuals. But then it is also stated that that the final sample (n = 110 044) is 81.4% of the original sample – is this % correct? (2) In the methodology, it is stated that the absolute risks and 95% confidence intervals are calculated, but the caption for Figure 2 states that the absolute risks and 99% confidence intervals are presented? Please standardise this. (3) Figure 2 – the y-axis needs to be labelled. (4) Please check the caption for Table 3 – is model 7 presented in Table 2 or 3? (5) It is stated in the methods that Cox proportional hazards regression with a constant follow-up time equal to one, is used to obtain prevalence ratios (PRs) with 95% CIs but then the relative risk (RR) is presented thereafter – are the results supposed to indicate the PRs or RRs? Or is the PR = RR? Or does the PR approximate to the relative risk? (if so, please clearly state in the methods). (6) It is stated that the discriminatory accuracy (DA) is quantified by means of the area under the receiver operator characteristics curve (AUC). And intersectional inequalities according to the DA is quantified as (i) “absent or very small” (AUC= 0.5–0.6), (ii) “moderate” (AUC >0.6–≤ 0.7), (iii) “large” (AUC >0.7– ≤ 0.8) and (iv) “very large” (AUC> 0.8) based on the classification provided by Hosmer and Lemeshow. Can this classification be further justified?, because according to the reference given, (AUC ≥0.7– < 0.8) is considered as acceptable discrimination, and above that is considered excellent etc. (7) For smoking status - what is the exact criterion distinguishing “Yes” or “Yes, sometimes” for the question “Do you smoke”? Was a sensitivity analysis done to see if there were significant differences between these two groups? (8) Potential strategies to reduce socioeconomic disparities between immigrants and natives are discussed. But very little
---

	information is given on these immigrants. The definition of “immigrant” in the study is implied as “not born in Sweden”. Can more information be provided on this cohort – e.g. major occupation sector, employment status, region of origin etc? (9) It is stated that native cohabitating males in the 30-44-year-old age group, with high educational achievement, was made the reference group as they are assumed to occupy the position of greatest structural privilege. Does “structural privilege” refer to concepts of economic, social, and cultural capital? Has this been discussed in previous work? Please elaborate or cite a reference at least. (10) It is stated that a sensitivity analysis was done using income as an indicator of SEP – can these results be presented in the supplementary material?
--	---

REVIEWER	Dr Martin Mlinaric Institute of Medical Sociology (IMS), Medical Faculty – Martin Luther University Halle-Wittenberg, Germany
REVIEW RETURNED	26-Oct-2020

GENERAL COMMENTS	I welcome and appreciate this study, as it displays intersectional patterns of smoking and within-differences based on an unique Swedish sample. However, I'm a bit afraid that some intersectional scholars, and especially those coming from qualitative backgrounds or “anti-categorical” intersectional researchers, might see more limitations and problems than the authors are currently ready to discuss. Quantitative studies need to create clear (sub-)groups to perform substantial statistical analyzed and may by this tend to generalize groups, which may result in "categorical fetishism" and "othering" of some groups. For instance, smoking status and migration background are assessed binary in this study, which is certainly not the best option to study heterogeneity within and across groups, but probably also a result of the conducted survey design which may limit the researchers to create more complex categories, intersectionality would plead for. I would avoid the term gender throughout the whole manuscript, as the authors apply sex (binary category male/female). Gender (identity) goes beyond (e.g., sexuality, LGBTIQ) the biological classifications of sex/gender that most quantitative studies are able to assess. In most quantitative surveys sex is assessed or at best a third option, but this not necessarily gender from the perspective of intersectionality. I'd furthermore suggest in engaging with the following literature which I miss in the references and discussion, as I believe that they are crucial to the field. What is the role of institutions/tobacco control and capitalism with regards to smoking inequalities for instance? How should we/should not incorporate intersectionality theory into population health research and health monitoring? See: Gkiouleka, A., Huijts, T., Beckfield, J., & Bambra, C. (2018). Understanding the micro and macro politics of health: Inequalities, intersectionality & institutions - A research agenda. Social Science & Medicine (1982), 200, 92–98. https://doi.org/10.1016/j.socscimed.2018.01.025
--

	Green, M. A., Evans, C. R., & Subramanian, S. V. (2017). Can intersectionality theory enrich population health research? Social Science & Medicine (1982), 178, 214–216. https://doi.org/10.1016/j.socscimed.2017.02.029 Bauer, G. R. (2014). Incorporating intersectionality theory into population health research methodology: challenges and the potential to advance health equity. Social Science & Medicine (1982), 110, 10–17. https://doi.org/10.1016/j.socscimed.2014.03.022 Bauer, G. R., & Scheim, A. I. (2018). Methods for analytic intercategory intersectionality in quantitative research: Discrimination as a mediator of health inequalities. Social Science & Medicine (1982). Advance online publication. https://doi.org/10.1016/j.socscimed.2018.12.015 Is this approach in the paper an intra, anti- or inter-categorical approach to intersectionality? Authors should contextualize and clarify their contribution to the field. See: McCall, L. (2005). The Complexity of Intersectionality. Signs: Journal of Women in Culture and Society, 30(3), 1771–1800. https://doi.org/10.1086/426800 What are the general limitations and challenges when studying with quantitative methods (applying intersectional analysis of individual heterogeneity and discriminatory accuracy (AIHDA)? Adequate representation of some (migrant or sex/gender) groups is a serious concern, especially from an intersectional perspective, as the sample should be at best well balanced with minority and majority groups (see Gkiouleka et al.). In this study, migration was dichotomized, which is problematic from the perspective of intersectionality theory. Ex-Yugoslav or Eastern European migrants of both genders for instance smoke more than Asian females or African males. Syrian refugees have on average higher educational levels than other refugee groups, which may also influence their respective smoking habits (healthy migrant effect). Finally, there are some qualitative intersectional studies in relation to smoking the authors should pay attention to: e.g., Triandafilidis, Z., Ussher, J. M., Perz, J., & Huppatz, K. An Intersectional Analysis of Women's Experiences of Smoking-Related Stigma. Qualitative Health Research
--	--

VERSION 1 – AUTHOR RESPONSE

Reviewer: 1
Rosemary Hiscock

Comments to the Author

R1.1: This study breaks the Swedish population into different sociodemographic groups and looks at changes in smoking prevalence over time.

Au: We would like to stress that our study does not primarily look at changes in smoking prevalence over time. Rather, we apply a cross-sectional approach analysing intersectional differences in the period prevalence adjusted for survey year. This is justified considering that the changes in gaps between the included socioeconomic and demographic categories are generally small.

R1.2: The study finds that not all low SES groups have high prevalence and not all high SES groups have low prevalence. I think this point could be made clearer by adding a table/figure comparing smoking rates by low, mid, high education for different sub groups.

Au: Yes, in other words, we show heterogeneity in the prevalence of smoking across many intersectional groups, which provides an improved map of the distribution of smoking in the population. In the figure 2 in the original manuscript the information requested by the referee can be found.

R1.3: Throughout:

“civil status” should be changed to “household composition” – otherwise could be about citizenship or marital status. Also ‘cohabiting’ normally refers to with a partner so please change to ‘living with other(s)’

Au: We follow the recommendations of the referee and in the revised manuscript we use “household composition” rather than “civil status” as well as “living with other(s)” rather than “cohabiting”.

R1.4: P6 line 32 to 39 taxes and smoke free are more effective than smoking cessation programs <https://link.springer.com/article/10.1007/s12254-019-0485-6>

Au: Thank you for this comment. The original phrasing that “smoking cessation campaigns may be a logical step” was not intended to express strong belief in their efficiency and we agree that structural interventions at a higher level are more effective both when it comes to reducing overall prevalence (which the cited article by Neuberger shows) and to reduce socioeconomic disparities. In the revised manuscript we incorporate the idea as well as the reference provided by the referee, see manuscript revision incorporated in the response to the following question.

R1.5: I also disagree that it necessarily widens inequalities – if it is targeted to areas with high smoking rates (low SES) then it can reduce inequalities <https://www.ncbi.nlm.nih.gov/pmc/articles/PMC2807194/>

Au: Thank you for this reference. Considering the modest effect in reducing socioeconomic disparities observed in the reference provided by the reviewer, we consider that even that publication supports the importance of proportionate universalism. Indeed, the launching of the NHS smoking cessation campaign constitutes an example of proportionate universalism since this intervention was first launched in areas with higher needs but is now available across the country. In the revised manuscript we incorporate the reference provided by the referee and clarify the conclusions of the reference by Östergren and Vilhelmsson originally cited:

p5, line 11:

“Structural interventions including raised tobacco taxes and smoking free zones can reduce smoking prevalence [1], most among people with low SEP [2]. In UK, health-care based smoking cessation aid has reduced disparities in smoking rates between privileged and socioeconomically deprived areas, although this effect was modest [3]. However, a review of the efficacy of non-health care interventions targeting behavioural factors among people with low education [4] concludes that there is a lack of evidence that such interventions oriented towards individual determinants of health are efficient when

it comes to reducing socioeconomic disparities in smoking [5].”

R1.6: P9 line 6 why exclude those under 30 years old?

Au: The lower age limit of 30 years was chosen since most individuals in Sweden have reached their highest educational achievement by this age and educational status was the indicator of socioeconomic position chosen in this study. This has been clarified in the revised manuscript.

p8, line 4:

“The lower age limit of 30 years was chosen since most individuals in Sweden that will complete a three-year education after high school do so before this age [6] and educational status was the indicator of socioeconomic position chosen in this study.”

R1.7: P9 line 25 I think you should make clearer here that education is a measure of SES and the other variables are demographic characteristics

Au: We have made this clear in the revised manuscript, see response to the previous question.

R1.8: P9 Line 56 add 'linked' before 'information'

Au: We have added this information in the revised manuscript.

R1.9: P10 line 35 Is this not just a forward stepwise?

Au: In forward stepwise regression, variable selection involves starting with no variables in the model, testing the addition of each variable using a chosen model fit criterion, adding the variable whose inclusion gives the most statistically significant improvement of the fit, and repeating this process until none improves the model to a statistically significant extent. We do not do so. We decide à priori which variables define the intersectional strata.

R1.10: P10 line 54 Why enter in that order?

Au: We started with age, sex and education status since these are categories that are most commonly reported in health reports in Sweden. The order after these variables is arbitrary and does not reflect any hierarchical importance given to the different variables.

R1.11: P12 line 27 “such differences were similar” would it be better to say “the gaps between subgroups were static”? - however looking at the graphs S1 – the gap for age categories declines over time, the gender with the highest smoking rate changes but there is little difference by gender. Would it be possible to include confidence intervals? Partos et al’s figure 2 shows a good way of including confidence intervals using shading <https://www.ncbi.nlm.nih.gov/pmc/articles/PMC5934656/> and here figure 11 <https://www.ncbi.nlm.nih.gov/books/NBK555625/>

Au: Our main objective in the present work was to study differences in the period prevalence of smoking. Therefore, we adjusted for survey year. However, we agree with the referee in that the gap for age categories declines over time and we have now modified the summary of the findings in S1-graph as described below and we have added 95% CI to the figures. In a coming study, we plan to investigate intersectional differences in the temporal change (reduction) of smoking risk.

p11 line 16:

While sex-differences were small throughout the period and the sex-category with highest smoking prevalence changed, we observed consistent differences between groups defined by age, country of birth, educational achievement and household composition. In absolute terms, the gaps between subgroups were static except for differences between age categories that narrowed in later years (see supplementary information S1).

Discussion

R1.12: P16 to 17 A big limitation and area for future work is that you lumped all foreign born participants together. Some ethnic groups/countries will have higher smoking rates than those born in Sweden and others will have lower (particularly if differentiated by gender). It would be helpful to include the largest ethnic minorities in Sweden and whether the smoking rates of these countries are higher than Sweden. Also the ethnic make up of your oldest age group is possibly different to the ethnic composition of your youngest age group?

Au: Yes, we agree with the referee, this question is a relevant area for future work. We agree in that the dichotomization of migration status is a limitation of this study. However, considering the size of the database and the information available on country of birth a more detailed classification only allows four categories (i.e., Sweden, Nordic countries, Europe and "others") This categorization is also used by the Swedish Public Health authority. Using this categorization we can see that all the categories except women born Outside Europe women have a higher prevalence than the individuals born in Sweden:

Region of birth	Women	Men
Sweden	17.5%	16.4%
Nordic countries	21.7%	23.0%
Europe	24.0%	27.1%
Outside Europe	16.3%	32.5%

In the revised manuscript, we provide this table as a supplementary material and add the following section to the discussion, to highlight that the dichotomization of migration status is a weakness:

p16 line 3:

"A further limitation of this study is the simple categorizations of the dimensions incorporated in the intersectional matrix. Gender was binary defined which neglects the existence of numerous gender identities. Migration status was binary defined as natives and immigrants, which may hide heterogeneity in smoking prevalence. A more detailed classification with four categories (i.e., Sweden, Nordic countries, Europe and "others") shows that all the categories except women born outside Europe have a higher prevalence than the individuals born in Sweden (see supplementary material 4). The used categorization stems in part from the information available in the survey and in part from the aim of presenting a parsimonious intersectional model that is easier to adopt in public health analyses and by the fact that several strata would be empty or contain very few individuals if the intersectional matrix was expanded."

R1.13: Figure 2 Please turn to landscape so that all the categories on the x axis can be seen

Au: We have modified the figure 2 according the indication of the referee.

Reviewer: 2

Jeevitha Mariapun

Comments to the Author

R2 (0) This is an interesting piece of work that uses a simplified methodology to describe intersectional patterns of smoking and quantify heterogeneities within groups in a population. There are a few comments as follows:

Au: Thank you very much for your positive evaluation.

R2(1) It is stated that the original sample has 136 301 individuals. But then it is also stated that that the final sample (n = 110 044) is 81.4% of the original sample – is this % correct?

Au: Thank you for noticing this mistake. The proportion of the original sample that is included in the final sample is 80.7% and we have corrected this in the revised manuscript. 81.4% referred to the proportion that were included prior to exclusion of N=850 people with missing data on smoking.

R2 (2) In the methodology, it is stated that the absolute risks and 95% confidence intervals are calculated, but the caption for Figure 2 states that the absolute risks and 99% confidence intervals are presented? Please standardise this.

Au: Thank you for this observation. It should be 95% CI. We have corrected this information.

R2 (3) Figure 2 – the y-axis needs to be labelled.

Au: Thank you for this observation. We have modified the y-axis to express percentages and included the appropriate label.

R2 (4) Please check the caption for Table 3 – is model 7 presented in Table 2 or 3?

Au: Thank you for this observation. We have checked and corrected the caption for Table 3. The model 7 is presented in table 2.

R2 (5) It is stated in the methods that Cox proportional hazards regression with a constant follow-up time equal to one, is used to obtain prevalence ratios (PRs) with 95% CIs but then the relative risk (RR) is presented thereafter – are the results supposed to indicate the PRs or RRs? Or is the PR = RR? Or does the PR approximate to the relative risk? (if so, please clearly state in the methods).

Au: Most properly the results indicate prevalence ratios (PR). We have consistently applied this denomination in the revised manuscript.

R2 (6) It is stated that the discriminatory accuracy (DA) is quantified by means of the area under the receiver operator characteristics curve (AUC). And intersectional inequalities according to the DA is quantified as (i) “absent or very small” (AUC= 0.5–0.6), (ii) “moderate” (AUC >0.6–≤ 0.7), (iii) “large” (AUC >0.7– ≤ 0.8) and (iv) “very large” (AUC> 0.8) based on the classification provided by Hosmer and Lemeshow. Can this classification be further justified?, because according to the reference given, (AUC ≥0.7– < 0.8) is considered as acceptable discrimination, and above that is considered excellent etc.

Au: Thank you for this remark. We follow the cut-off provided by Hosmer and Lemeshow but we prefer using adjectives like “large” rather than “acceptable” as “large” is a neutral qualification. In the

revised manuscript we have clearly stated that it is the cut-off-values we follow and that we use more neutral denominations:

p10 line 24:

“However, based on the cut-off values provided by Hosmer and Lemeshow [7] but using more neutral denominations we qualify intersectional inequalities according to the DA as (i) “absent or very small” (AUC= 0.5–0.6), (ii) “moderate” (AUC >0.6–≤ 0.7), (iii) “large” (AUC >0.7– ≤ 0.8) and (iv) “very large” (AUC >0.8).”

R2 (7) For smoking status - what is the exact criterion distinguishing “Yes” or “Yes, sometimes” for the question “Do you smoke”? Was a sensitivity analysis done to see if there were significant differences between these two groups?

Au: The exact questions have undergone minor changes between 2004 and 2018. Throughout 2004-2015 respondents were asked “do you smoke every day” and if they reported not smoking everyday they were subsequently asked whether they “smoked every now and then” and “have you previously smoked every day during at least 6 months”? Since 2016, people were asked the question “Do you smoke” and could choose between a) “Yes, every day” → how many cigarettes per day? b) “Yes, sometimes” and c) No. People that answered “Yes” or “Yes, sometimes in 2016 and 2018” and people that answered that they smoked “every day” and “every now and then” between 2004 and 2015 were considered as smokers. Previous smoking status was not assessed in this study.

In a sensitivity analysis where we restricted the outcome to include only everyday smokers, the overall prevalence of smoking dropped from 18% to 11.37%. The à priori designated reference group had the lowest prevalence of 1.6% compared to 11.8% in the original analysis. The intersectional disparities were also larger with prevalence ratios ranging from 1 (reference group) to 26.16 (16.95-40.39) for young native men with low education that live alone and the AUC of model 7 was 0.70 with everyday smoking as the outcome compared to 0.66 when sometimes smokers were included.

Our main results combined with this sensitivity analysis show that intersectional disparities exist not only for smoking prevalence but also for intensity in smoking. Our results could thus underestimate intersectional disparities in health hazards attributable to smoking. We have added a section in the discussion where we highlight this:

p16 line 14:

“We also performed a sensitivity analysis excluding “sometimes smokers” from the smoker category. As expected, overall prevalence was lower, 11% compared to 18%, and intersectional disparities larger. The AUC of the intersectional model 7 was 0.70 compared to 0.66 in the main analysis. Our main results combined with the results from the sensitivity analysis reflect the existence of socioeconomic disparities not only in prevalence, but also in intensity, of smoking [8]. Our results therefore may underestimate the intersectional disparities in health hazards attributable to smoking.”

R2 (8) Potential strategies to reduce socioeconomic disparities between immigrants and natives are discussed. But very little information is given on these immigrants. The definition of “immigrant” in the study is implied as “not born in Sweden”. Can more information be provided on this cohort – e.g. major occupation sector, employment status, region of origin etc?

Au: Please see the response to reviewer 1.12 that raised a similar concern. As the reviewer remarks, other aspects of SEP such as employment status and occupation sector are relevant and the heterogeneity within the broad category “not born in Sweden” is certainly large. The aim of this study

is to present a simple intersectional model that can be broadly implemented and this requires trade-offs regarding how nuanced each dimension can be categorized. In our database, inclusion of a more detailed categorization of migration status would result in many empty strata or strata with very few individuals.

R2 (9) It is stated that native cohabitating males in the 30-44-year-old age group, with high educational achievement, was made the reference group as they are assumed to occupy the position of greatest structural privilege. Does “structural privilege” refer to concepts of economic, social, and cultural capital? Has this been discussed in previous work? Please elaborate or cite a reference at least.

Au: Thank you for the opportunity to clarify the choice of reference group. Intersectionality aims at exploring the simultaneous effect of being subjected to different processes of structural oppression, e.g. sexism, racism, heteronormativity, capitalism and others. Although the dynamics of these oppressive processes change between places and times, men are generally considered privileged compared to women, people with high socioeconomic status (SES) privileged compared to low SES, natives privileged compared to immigrants and people living together with others privileged compared to people living alone. Ageism is not as frequently focused in intersectional research but may contribute to worse health among elderly [9]. Therefore, the youngest age category was indicated as a reference group. In addition to this, the cohort effect on smoking (i.e. the different susceptibility to take up smoking depending on the period in which an individual lives) makes it important to consider age.

Usually, intersectionality scholars do not explicitly give preference to any specific form of capital. Rather, inequities in access to both economic, social, cultural as well as symbolic capital is implicitly indicated as mechanisms contributing to intersectional health inequities. Different sorts of capital may be differently important for the different dimensions in an intersectional matrix. For example, economic and cultural capital may be most important for SES differences in health, social capital contributes to health differences between immigrants and natives. Taking educational achievement as an indicator of cultural capital and maintaining the binary assessment of sex, women in Sweden today have higher cultural capital than men but have less economic capital.

In conclusion, we cannot state that structural privilege refers to any form of capital but have justified the choice of reference group with references in the following section:

p9 line 8:

“We used 30–45-year-old, native men living with other(s) and with high educational achievement as the reference in the comparisons, as this group was assumed to occupy the position of greatest structural privilege. This choice was based on unidimensional assumptions of structural privilege for young compared to old [9], men compared to women, high SEP compared to low SEP [10], natives compared to immigrants [11] and those living with other(s) compared to people living alone [12].”

R2 (10) It is stated that a sensitivity analysis was done using income as an indicator of SEP – can these results be presented in the supplementary material?

Au: We have added those results in the supplementary material, see supplementary material 3.

Reviewer: 3

Dr Martin Mlinaric

Comments to the Author

R3.1 I welcome and appreciate this study, as it displays intersectional patterns of smoking and within-differences based on an unique Swedish sample.

Au: Thank you very much for your positive evaluation.

R3.2: However, I'm a bit afraid that some intersectional scholars, and especially those coming from qualitative backgrounds or "anti-categorical" intersectional researchers, might see more limitations and problems than the authors are currently ready to discuss. Quantitative studies need to create clear (sub-)groups to perform substantial statistical analyzed and may by this tend to generalize groups, which may result in "categorical fetishism" and "othering" of some groups. For instance, smoking status and migration background are assessed binary in this study, which is certainly not the best option to study heterogeneity within and across groups, but probably also a result of the conducted survey design which may limit the researchers to create more complex categories, intersectionality would plead for.

Au: We agree with the referee that quantitative studies necessarily simplify complex categories, and are aware that from an anti-categorical point of view studies like this one may be considered to perpetuate inequities by using categorizations that are overly simplified and reinforce stigma. Merlo has discussed the tension between anti- and inter-categorical intersectionality in previous publications [13-15]. We believe that this AIHDA-approach is suitable to determine whether an anti- or inter-categorical intersectionality is most appropriate for a given outcome in a given context. Specifically, the evaluation of the discriminatory accuracy of an intersectional model provides information that can guide such judgement. If the DA is very low, that supports the anti-categorical intersectionality since the categories used do not to improve the understanding of smoking patterns in Sweden. If the DA is large, the inter-categorical intersectionality becomes more relevant since the intersectional strata, despite remaining heterogeneity, indeed help us understand which intersectional groups are at increased risk of smoking.

The binary assessment of migration status was applied since with a more nuanced categorization the number of strata would increase rendering empty strata or strata with very few individuals.

We have incorporated a section where we elaborate how AIHDA may serve to guide the choice between an anti- or inter-categorical intersectionality.

P6 line 22:

"A low DA suggests the need for universal interventions while a high DA supports more selective interventions. This idea aligns with the distinctions made by McCall between anti-, intra- and inter-categorical intersectional approaches [16]. According to the anti-categorical intersectionality, the categorizations adopted in quantitative research are overly simplified and contribute to stereotypes and perpetuations of inequalities. The inter-categorical intersectionality, on the other hand, accepts imperfect categorizations since they can be useful in the study of intersectional inequities. The finding of a low DA would support the anti-categorical standpoint that the categorizations lack relevance for the studied outcome. If the DA is high, this would rather support the inter-categorical standpoint that intersectional matrix provides worthy information. A moderate DA does not give full support to neither the anti- nor inter-categorical intersectionality."

We have also highlighted that the binary assessment of some variables is a limitation. See our answer to another referee (R1.12).

R3.3 I would avoid the term gender throughout the whole manuscript, as the authors apply sex (binary category male/female). Gender (identity) goes beyond (e.g., sexuality, LGBTIQ) the biological classifications of sex/gender that most quantitative studies are able to assess. In most quantitative surveys sex is assessed or at best a third option, but this not necessarily gender from the perspective of intersectionality.

Au: We agree with the referee that the binary assessment of male/female is a limitation, we highlight this in the revised manuscript. Both for gender and biological sex, the dichotomization is a simplification since intersex categories outside the dichotomization of biological sexes also exist. In the present study, we choose the determination gender since we believe that the (changing) smoking patterns between men and women are due to socially construct of gender rather than biological sex.

R3.4 I'd furthermore suggest in engaging with the following literature which I miss in the references and discussion, as I believe that they are crucial to the field. What is the role of institutions/tobacco control and capitalism with regards to smoking inequalities for instance? How should we/should not incorporate intersectionality theory into population health research and health monitoring?

See:

Gkiouleka, A., Huijts, T., Beckfield, J., & Bambra, C. (2018). Understanding the micro and macro poliics of health: Inequalities, intersectionality & institutions - A research agenda. *Social Science & Medicine* (1982), 200, 92–98. <https://doi.org/10.1016/j.socscimed.2018.01.025>

Green, M. A., Evans, C. R., & Subramanian, S. V. (2017). Can intersectionality theory enrich population health research? *Social Science & Medicine* (1982), 178, 214–216. <https://doi.org/10.1016/j.socscimed.2017.02.029>

Bauer, G. R. (2014). Incorporating intersectionality theory into population health research methodology: challenges and the potential to advance health equity. *Social Science & Medicine* (1982), 110, 10–17. <https://doi.org/10.1016/j.socscimed.2014.03.022>

Bauer, G. R., & Scheim, A. I. (2018). Methods for analytic intercategory intersectionality in quantitative research: Discrimination as a mediator of health inequalities. *Social Science & Medicine* (1982). Advance online publication. <https://doi.org/10.1016/j.socscimed.2018.12.015>

Au: Thank you for this comment and for literature provided. While Bauer (2014) was already included in the literature referenced we have now added the suggested publication by Green et al (2017) and the publication by Gkiouleka (2018). However, while the study on discrimination as a mediator of health inequalities certainly is relevant for the field and we hope to see more studies assessing causal processes generating intersectional disparities, that discussion falls outside the main focus of this study.

We agree with Gkiouleka et al. that an intersectional perspective should be combined with the study of the role of institutions as determinants of health, in order not to focus solely on individuals. We have expanded our discussion section in order to clarify both our view on the role of the institutions and how capitalism may affect smoking inequalities.

p17 line22:

“Equal access to education, housing and healthy recreation, regardless of gender, socioeconomic status, migration status and household composition, is important to reduce smoking prevalence. Therefore, institutions outside the health care system play an important role to redistribute resources and access to SDH [17, 18], in order to counterweight the accelerating tendency of accumulation of resources among a very rich minority that characterizes modern capitalism [19]. This requires political

decisions that prioritize population health aims more than market oriented reforms that exacerbate health inequities [20]. Health politics should adopt an intersectional perspective when redistributing resources in order to reduce the complex disparities in smoking revealed in this study.”

R3.5 Is this approach in the paper an intra, anti- or inter-categorical approach to intersectionality? Authors should contextualize and clarify their contribution to the field.

See: McCall, L. (2005). The Complexity of Intersectionality. *Signs: Journal of Women in Culture and Society*, 30(3), 1771–1800. <https://doi.org/10.1086/426800>

Au: À priori we did not decide whether to adopt an anti-, intra- or inter-categorical intersectional perspective, as explained in the section included as a response to R3.2. Based on our findings, we cannot fully support neither the anti- nor the inter-categorical approach. The following section was added to the discussion:

p13 line 15:

We found a moderate AUC= 0.66, which indicates that individual risk of smoking considerably overlaps between the intersectional strata and that neither the anti-categorical nor the inter-categorical intersectionality approaches are fully supported.

R3.6 What are the general limitations and challenges when studying with quantitative methods (applying intersectional analysis of individual heterogeneity and discriminatory accuracy (AIHDA)? Adequate representation of some (migrant or sex/gender) groups is a serious concern, especially from an intersectional perspective, as the sample should be at best well balanced with minority and majority groups (see Gkiouleka et al.). In this study, migration was dichotomized, which is problematic from the perspective of intersectionality theory. Ex-Yugoslav or Eastern European migrants of both genders for instance smoke more than Asian females or African males. Syrian refugees have on average higher educational levels than other refugee groups, which may also influence their respective smoking habits (healthy migrant effect).

Au: We agree with the reviewer that the dichotomization of migration status is a limitation. The problem of crude categorizations is an inherent problem to quantitative intersectional research aiming at presenting models that are sufficiently simplified to be useful for regular health monitoring outside academia. See also our answer to another referee (R1.12), In any case, an advantage of AIHDA is that the assessment of Discriminatory Accuracy transparently shows the heterogeneity that remains unexplained and thus cautions against too far-reaching conclusions.

R3.7 Finally, there are some qualitative intersectional studies in relation to smoking the authors should pay attention to:

e.g., Triandafilidis, Z., Ussher, J. M., Perz, J., & Huppatz, K. An Intersectional Analysis of Women’s Experiences of Smoking-Related Stigma. *Qualitative Health Research*

Au: Thank you for this literature advice that we had not read before. We have expanded the discussion on implications with the following section, citing Triandafilidis et al as well as Graham [21]:

p17 line 15:

Interventions to reduce smoking prevalence should address Social Determinants of Health (SDH) at all levels. Examples targeted directly at smoking include increased tobacco taxation, smoke free zones and public anti-smoking campaigns [22]. Stigmatization is a negative side effect of such

interventions that need to be taken into account, especially for low SEP groups [21]. Qualitative intersectional research has provided important insights into how the stigma of smoking interacts with identities of low class, country of birth, being a bad mother and may be in conflict with norms of femininity [23].

REFERENCES

- [1] Neuberger M. Tobacco control: prevention and cessation in Europe. *memo - Magazine of European Medical Oncology* 2019;12(2):156-161.
- [2] Golden SD, Farrelly MC, Luke DA, et al. Comparing projected impacts of cigarette floor price and excise tax policies on socioeconomic disparities in smoking. *Tobacco control* 2016;25(Suppl 1):i60.
- [3] Bauld L, Judge K, Platt S. Assessing the impact of smoking cessation services on reducing health inequalities in England: observational study. *Tobacco control* 2007;16(6):400-404.
- [4] Vilhelmsson A, Ostergren PO. Reducing health inequalities with interventions targeting behavioral factors among individuals with low levels of education - A rapid review. *PLoS one* 2018;13(4):e0195774.
- [5] Wendel-Vos GC, Dutman AE, Verschuren WM, et al. Lifestyle factors of a five-year community-intervention program: the Hartsлаг Limburg intervention. *American journal of preventive medicine* 2009;37(1):50-56.
- [6] OECD. Tertiary-type A graduation rates in 1995, 2000 and 2008 (first-time graduation) 2010.
- [7] Hosmer DW, Lemeshow S. *Applied logistic regression*. 2nd ed. New York: Wiley 2000.
- [8] Bobak M, Jarvis MJ, Skodova Z, et al. Smoke intake among smokers is higher in lower socioeconomic groups. *Tobacco control* 2000;9(3):310.
- [9] McMullin JA, Cairney J. Self-esteem and the intersection of age, class, and gender. *Journal of Aging Studies* 2004;18(1):75-90.
- [10] Krieger N. A glossary for social epidemiology. *Journal of epidemiology and community health* 2001;55(10):693.
- [11] Karlsen S, Nazroo JY. Measuring and analyzing "race," racism, and racial discrimination. *Methods in social epidemiology* 2017;2:43.
- [12] Alm S, Nelson K, Nieuwenhuis R. The Diminishing Power of One? Welfare State Retrenchment and Rising Poverty of Single-Adult Households in Sweden 1988–2011. *European Sociological Review* 2019;36(2):198-217.
- [13] Wemrell M, Mulinari S, Merlo J. Intersectionality and risk for ischemic heart disease in Sweden: Categorical and anti-categorical approaches. *Social science & medicine* (1982) 2017;177:213-222.
- [14] Mulinari S, Bredstrom A, Merlo J. Questioning the discriminatory accuracy of broad migrant categories in public health: self-rated health in Sweden. *European journal of public health* 2015.
- [15] Axelsson Fisk S, Mulinari S, Wemrell M, et al. Chronic Obstructive Pulmonary Disease in Sweden: an intersectional multilevel analysis of individual heterogeneity and discriminatory accuracy. *SSM-Population Health* 2018.
- [16] McCall L. The complexity of intersectionality. *Signs: Journal of women in culture and society* 2005;30(3):1771-1800.
- [17] Gkiouleka A, Huijts T, Beckfield J, et al. Understanding the micro and macro politics of health: Inequalities, intersectionality & institutions-A research agenda. *Social Science & Medicine* 2018;200:92-98.
- [18] Bambra C, Fox D, Scott-Samuel A. Towards a politics of health. *Health promotion international* 2005;20(2):187-193.
- [19] Piketty T. About capital in the twenty-first century. *American Economic Review* 2015;105(5):48-53.
- [20] Burström B. Market-Oriented, Demand-Driven Health Care Reforms and Equity in Health and Health Care Utilization in Sweden. *International Journal of Health Services* 2009;39(2):271-285.
- [21] Graham H. Smoking, Stigma and Social Class. *Journal of Social Policy* 2011;41(1):83-99.
- [22] Dahlgren G, Whitehead M. *Policies and strategies to promote social equity in health*. 1991.

[23] Triandafilidis Z, Ussher JM, Perz J, et al. An Intersectional Analysis of Women's Experiences of Smoking-Related Stigma. *Qualitative Health Research* 2016;27(10):1445-1460.

VERSION 2 – REVIEW

REVIEWER	Rosemary Hiscock University of Bath UK
REVIEW RETURNED	21-Dec-2020
GENERAL COMMENTS	The authors' responses have satisfied my concerns
REVIEWER	Dr. Jeevitha Mariapun Clinical School Johor Bahru, Jeffrey Cheah School of Medicine and Health Sciences, Monash University Malaysia
REVIEW RETURNED	05-Jan-2021
GENERAL COMMENTS	This is an interesting piece of work that uses a simplified methodology to describe intersectional patterns of smoking and quantify heterogeneities within groups in a population. Thank you, all comments from the first review have been addressed satisfactorily. The only other comment would be that the authors double-check the labeling/caption of tables, e.g., in S3, PRs are labeled as RRs.